# Imaging Biomarkers for Monitoring the Inflammatory Redox Landscape in the Brain

**DOI:** 10.3390/antiox10040528

**Published:** 2021-03-28

**Authors:** Eduardo Felipe Alves Fernandes, Dennis Özcelik

**Affiliations:** 1Department of Drug Design and Pharmacology, University of Copenhagen, Universitetsparken 2, 2100 Copenhagen, Denmark; eduardo.fernandes@sund.ku.dk; 2Chemistry | Biology | Pharmacy Information Center, ETH Zürich, Vladimir-Prelog-Weg 10, 8093 Zürich, Switzerland

**Keywords:** oxidative stress response, reactive oxygen species, imaging biomarker, redox sensor, microglia, positron emission tomography, TSPO, MAO–B

## Abstract

Inflammation is one key process in driving cellular redox homeostasis toward oxidative stress, which perpetuates inflammation. In the brain, this interplay results in a vicious cycle of cell death, the loss of neurons, and leakage of the blood–brain barrier. Hence, the neuroinflammatory response fuels the development of acute and chronic inflammatory diseases. Interrogation of the interplay between inflammation, oxidative stress, and cell death in neurological tissue in vivo is very challenging. The complexity of the underlying biological process and the fragility of the brain limit our understanding of the cause and the adequate diagnostics of neuroinflammatory diseases. In recent years, advancements in the development of molecular imaging agents addressed this limitation and enabled imaging of biomarkers of neuroinflammation in the brain. Notable redox biomarkers for imaging with positron emission tomography (PET) tracers are the 18 kDa translocator protein (TSPO) and monoamine oxygenase B (MAO–B). These findings and achievements offer the opportunity for novel diagnostic applications and therapeutic strategies. This review summarizes experimental as well as established pharmaceutical and biotechnological tools for imaging the inflammatory redox landscape in the brain, and provides a glimpse into future applications.

## 1. Introduction

Neuroinflammation is a complex and poorly understood phenomenon with high clinical relevance. A large number of disorders of the central nervous system (CNS) show a substantial neuroinflammatory component. Notable examples are neuropsychiatric diseases, such as schizophrenia and bipolar disorder [1,2,3]. Inflammatory processes also play an important role in many autoimmune disorders of the CNS, of which multiple sclerosis (MS) is the most common one [4]. Despite the rapidly increasing incidence, the cause of MS is still under debate [5]. It is clear, however, that the immune system eliminates the myelin layers of nerve fibers [6]. These continuous attacks result in local inflammation sites within the myelin sheath and, eventually, recognizable lesions in magnetic resonance imaging (MRI) scans. These inflammation events disrupt the neuronal networks and, hence, are responsible for a highly diverse array of symptoms. Neurodegenerative diseases are another group of CNS disorders that display chronic inflammation of the brain. A notable representative of this group is Alzheimer’s disease (AD), which is also the leading cause of dementia [7].

A growing body of research provides evidence for chronic neuroinflammation as a cause for AD development, which is comprehensively summarized elsewhere [8,9]. For example, a number of studies found an elevated level of inflammation markers, e.g., the cytokines interleukin-1 beta (IL-1β), IL-6, and tumor necrosis factor α (TNF-α), in mouse models and AD patients [8,10,11,12]. Another study demonstrated that the innate immunity protein interferon-induced transmembrane protein 3 (IFITM3) is associated with Aβ production [13]. This study also linked the age-dependent increase of inflammatory markers, such as type I interferons [14], to concomitantly increased IFITM3 levels in the elderly, ultimately matching the clinical manifestation of AD as a late-developing neurological disease. Other studies showed that anti-inflammatory molecules enhance [15], or that proinflammatory molecules decrease Aβ clearance by the triggering receptor expressed on myeloid cells 2 (TREM-2) [16,17,18]. We refer the interested reader to the up-to-date review from Shi and Holtzman on the role of TREM-2, APOE, and disease-associated microglia in inflammation in AD [19].

In contrast to AD and other neurodegenerative diseases, which are associated with chronic inflammation, ischemic stroke is a representative example of a pathology that is associated with acute inflammation [20]. In ischemic stroke, occlusion of, usually, the middle cerebral artery reduces or blocks the blood flow in the human brain, which causes hypoxia in cortical brain areas [21]. Low oxygen levels trigger a vast and complex biochemical cascade encompassing inflammation, which finally causes neuronal cell death and impaired brain function [22].

A major factor for neuronal cell death in inflammation is oxidative stress. Oxidative stress is characterized by a relative increase of reactive oxygen species (ROS), which leads to an imbalance of the tightly regulated redox homeostasis of the cell [23,24]. This imbalance jeopardizes cellular function and survival, because ROS can react with a wide variety of biological biomolecules, causing, for instance, DNA damage [25] and protein modification [26], and eventually leading to apoptosis [27]. Moreover, ROS and oxidative stress can activate inflammatory signaling pathways and, hence, induce the production of numerous proinflammatory cytokines that drive inflammation [28]. Proinflammatory cytokines activate immune cells, amplifying both inflammation and ROS generation. Hence, the neuroinflammatory response fuels oxidative stress, which perpetuates inflammation.

Initial causes of the inflammatory responses and oxidative stress are perturbation in the cellular metabolism due to genetic anomalies or environmental factors. Typical examples of such environmental factors comprise lifestyle choices, for instance, physical exercise, diet, smoking, and alcohol consumption [29,30]. Consequently, these factors correlate with the development of stroke [31,32,33], neurodegenerative disorders [34,35], and other inflammatory brain diseases [36,37]. Other known external factors are pathogens, e.g., viruses and bacteria [38], and physical injuries, e.g., traumatic brain injury [39]. Environmental pollutants are another important factor that is connected to inflammation [40]. Many pollutants, e.g., metals [41], pesticides [42,43,44], and microplastic [45,46], are suspected to impair well-being. Furthermore, a large amount of research has established over the years that air pollution is associated with inflammatory processes in the body, including the brain [47,48,49]. For instance, a very recent study showed a correlation between air quality and the development of neurodegeneration [50].

In order to diagnose and treat many CNS diseases, it is crucial to understand, assess, and monitor the neuroinflammatory response in the brain. This, however, is a major challenge because of the complexity of the underlying biochemical processes of inflammation, particularly in the brain. Consequently, a lack of tools prohibits interrogating neuroinflammatory components and pathways for diagnostic and clinical purposes. The advent of chemical and synthetic biology might provide the research community and clinical professionals with novel means to trace the development of inflammatory processes in the human brain.

In this review, we summarize the inflammatory response and its interaction with the redox landscape, in particular, in the CNS. We further describe available and experimental tools and methods for monitoring key players in the redox landscape of inflammation in the brain. Finally, we present examples of promising applications in the clinical setting.

## 2. Redox Biology of Inflammation

### 2.1. Interplay of Oxidative Stress and Inflammation

Inflammation is a complex physiological process, which is associated with five characteristic symptoms: pain, redness, swelling, heat, and loss of function. Typically, inflammatory processes can be classified as acute or chronic inflammation (Figure 1). Acute inflammation is a direct response of the body and fades within a limited amount of time, usually within a few days. Subacute inflammation persists up to six weeks, whereas longer periods are deemed chronic. Chronic inflammation can last for years, and is associated with numerous pathologies (Figure 1).

Upon exposure to a pathogen, an irritant, or cell damage, leukocytes enter the affected tissue in order to eliminate the intruding pathogen, limit propagation of damage, and stimulate tissue repair processes [51]. Cells in the surrounding area secrete cytokines, in particular, IL-1β, IL-6, and TNF-α, to further attract and to activate cells of the immune system from the bloodstream or the local tissue [52,53]. An important effect of the inflammatory response is the alteration of redox environment [54,55].

The redox environment is tightly regulated to enable controlled biochemical reactions and molecular interactions in the cell and the organism [23,56,57,58]. In this environment, oxygen very often takes the prominent role as the terminal electron acceptor. The electron transfer to oxygen results in by-products, in particular, ROS that include superoxide anion (O_2_^•−^), hydroxyl radical (OH^•^), and hydrogen peroxide (H_2_O_2_). Naturally, cells produce a basal level of ROS in the mitochondria, the endoplasmic reticulum (ER), and the peroxisomes [59]. The major sources of intracellular ROS in these organelles are nicotinamide adenine dinucleotide phosphate (NADPH) oxidases (NOXs), mitochondrial electron transport chains, as well as ER oxidoreductases [60,61,62]. Under normal conditions, the cell is able to contain ROS, but overproduction leads to oxidative stress that impairs other physiological pathways [24,63].

A large number of studies established the interlocked nature of inflammation and oxidative stress. For instance, ROS regulate T cell responses, which indicates that oxidative stress can trigger an erroneous immune response, which fuels ongoing immune reactions [64,65]. In addition, ROS react with redox-sensitive proteins in the cell, and thus modulate crucial inflammatory signaling pathways, such as the MAPK/ERK pathway [66,67,68] or the NF-κB pathway [69,70,71]. Another important example of the interplay of oxidative stress and inflammation is the induction of the “respiratory burst” in phagocytic cells of the immune system, e.g., neutrophils, monocytes, and macrophages. This rapid production of ROS and other radicals, such as nitric oxide (NO^•^), serves to eliminate the pathogen and to stimulate the cytokine response by the leukocytes [72,73]. The central component of the increased ROS production is the NADPH oxidase 2, which is also termed NOX2 in humans [74,75]. NOX2 is present in multiple cell types and is responsible for the production of constitutive but low levels of ROS. During the respiratory burst, however, NOX2 activity increases substantially in phagocytes, which is associated with an up to 100-fold increase in oxygen consumption, and causes the physiologically induced oxidative stress event [76,77].

A large body of research provides compelling evidence for the important contribution of both oxidative stress and inflammation to the development of pathogen-independent inflammatory diseases, such as cancer, diabetes, or neurodegenerative diseases [78,79]. If left unchecked, inflammation and oxidative stress mutually perpetuate each other, driving the affected tissue in a vicious cycle to massive damage. Hence, the interplay of the redox landscape and inflammation draws considerable attention in current research efforts.

### 2.2. Induction of the Inflammatory Response

Activation of inflammatory signaling pathways is facilitated by various signals that also include oxidative stress. Upon recognition of an external stimulus, the innate immune system induces an inflammatory response. This recognition is mediated by pattern recognition receptors (PRRs) that sense pathogen-associated molecular patterns (PAMPs) and danger-associated molecular patterns (DAMPs). Besides membrane-bound PPRs, e.g., the Toll-like receptors (TLRs), that sense extracellular or endosomal signals, intracellular PPRs, including the nucleotide-binding domain and leucine-rich repeat-containing receptors (NLRs), and absent in melanoma 2 (AIM2)-like receptors (ALRs), recognize intracellular signals. A subgroup of cytosolic PPRs contributes to the assembly of the inflammasome [80]. The inflammasome is an intracellular multiprotein complex that activates and secretes the proinflammatory cytokines IL-1b and IL-18 [81]. Inflammasome activation depends on a number of factors, and ROS are known modulators of inflammasome activity [82].

It has been shown that H_2_O_2_ produced by the mitochondrial enzyme monoamine oxidase B (MAO–B) is crucially involved in the activation of the inflammasome [83]. This remarkable study also suggested that MAO–B could be exploited as a target for therapeutic modulation of inflammatory signaling. Besides direct modulation of inflammasomal activity, redox signaling can also indirectly regulate the inflammasome. For instance, an earlier study showed that liposomes could activate the NLRP3 inflammasome, which involves ROS and ROS-dependent calcium influx [84]. A more recent study showed that the 18 kDa translocator protein (TSPO), located in the outer mitochondrial membrane, is a key regulator of calcium homeostasis [85]. Since calcium signaling and mitochondrial destabilization induce the NLRP3 inflammasome [86], it seems to be plausible that TSPO contributes to the regulation of inflammasomal activity.

The activity of the inflammasome induces an inflammatory cascade that results in pyroptosis, a certain type of programmed cell death [87]. Pyroptosis requires activation of caspases, e.g., caspase-1, and stimulates the production of proinflammatory cytokines and gasdermins [88]. Gasdermins are pore-forming proteins that rupture the cell membrane, enabling the release of cytokines and DAMPs that perpetuate the inflammatory response by recruiting immune cells to the tissue [89,90]. Hence, inflammasomes play a key role in the inflammatory response, and were initially studied in cells of the innate immune system. PPRs are expressed in macrophages, NK cells, neutrophils, mast cells, and other cells of the innate immune system. Nevertheless, a large body of research also described inflammasomes in epithelial cells [80].

Given the central role in the inflammatory response, the inflammasome has drawn considerable attention as a drug target to tackle inflammatory diseases [91]. Inflammasome activation and cytokines production have been targeted by antagonists of the IL-1 receptor (e.g., anakinra), IL-1b neutralizing antibodies (e.g., canakinumab), and soluble decoy receptors for IL-1b and IL-1a (e.g., rilonacept) [92]. In addition, several compounds have been described for targeting subunits of the inflammasomes, such as NLRP3, caspase-1, and gasdermin D [93,94]. Despite all efforts, our current understanding of inflammasome signaling and the inflammatory response is still very limited, particularly in CNS inflammation.

### 2.3. The Neuroinflammatory Response

The brain, but also the entire central nervous system (CNS), is an immunologically privileged site, and possesses a distinct immune system that protects neuronal cells against pathogens [95,96]. The neuroimmune system comprises the highly diverse group of nonneuronal glial cells, which can roughly be divided into macroglia (e.g., astrocytes, enteric glial cells, ependymal cells, oligodendrocytes, radial glia, satellite cells, and Schwann cells) and microglia [97]. The latter ones are the major components of the neuroimmune system, and act as resident macrophages in the CNS [98]. They also express PPRs [99]; however, inflammatory signaling was also described for neurons [100], astrocytes [101], perivascular CNS macrophages [102], oligodendrocytes [103], and endothelial cells [104].

Upon activation, microglia release proinflammatory cytokines, such as IL-1β, IL-6, and TNF-α [52,105,106]. Like the immune cells of the bloodstream, microglia are also capable of undergoing respiratory burst phases that are mediated by an increase in NOX2 activity. ROS production is a crucial tool that microglia utilize in response to pathogens, protein aggregates, or cell debris [107,108]. For more detailed information on ROS generation in microglia, the interested reader is referred to a comprehensive review by Simpson and Oliver [109].

The inflammatory response in the brain and elsewhere is tightly regulated. Limited inflammation can fail to clear an infection, whereas overreaction can induce massive tissue damage [51,110,111]. Similarly, ongoing and chronic inflammation lead to severe side effects and physiological complications. A notable consequence of neuroinflammation affects the blood–brain barrier (BBB) [112]. The BBB consists of a layer of endothelial cells that create a highly selective semipermeable boundary, which separates the CNS from the circulating blood, and protects it from pathogens [113]. The BBB also prevents the passage of antibodies and immune cells. Moreover, the BBB limits the uptake of certain drugs, including antibiotics, to the CNS, making the BBB a considerable obstacle in the treatment of CNS diseases. Neuroinflammation can lead to increased permeability of the BBB, enabling infiltration of leukocytes, which can then participate in perpetuating the inflammatory response [51,111]. An overview of the neuroinflammatory response is presented in Figure 2.

Proinflammatory cytokines directly and indirectly increase neuronal cell damage. For instance, IL-1β causes neurotoxic effects by a variety of pathways [114], including the induction of oxidative stress [115,116]. IL-6 has an important function in neuronal tissue homeostasis, but overproduction leads to neurodegeneration [117,118]. Furthermore, IL-6 also mediates upregulation of VCAM-1 that leads to permeability of the BBB, allowing infiltration of leukocytes [119]. TNF-α recruits leukocytes from the periphery into the brain, and directly affects endothelial cells, eventually leading to increased BBB permeability [120]. In addition, TNF-α potentiates glutamate-induced cell death in hippocampal slides: it inhibits glutamate transporters, increases the expression of α-amino-3-hydroxy-5-methyl-4-isoxazolepropionic acid (AMPA) receptors, and, ultimately, elevates the levels of the oxidative effector nitric oxide synthase, which yields oxidative stress [121].

As illustrated, the redox landscape plays a crucial role in the development and progression of neurological diseases. Hence, experimental and translational methods are necessary to interrogate the progression of inflammatory and oxidative biomarkers in neurological diseases.

## 3. Tools and Methods to Monitor the Redox Landscape of Neuroinflammation

### 3.1. Monitoring the Redox Landscape in the Laboratory

Since inflammation and oxidative stress are tightly connected, the monitoring of ROS can provide valuable information about the biological processes during inflammation. One major challenge in this area of research is the limitation of available tools to monitor and to detect oxidative stress and inflammation. Over the years, scientists have developed a plethora of tools to study the production of ROS in cellular models. In general, there are two types of sensors to monitor the redox state in living cellular systems: genetically encoded redox proteins and small-molecule probes. These tools have been described numerous times, and we recommend a selection of comprehensive overviews that provide a detailed summary of genetically encoded redox proteins [122] and small-molecule redox sensors, including recent positron emission tomography (PET) tracers [123,124].

A significant body of research is devoted to the development and improvement of genetically encoded redox proteins. Recent advances led to novel fluorophores, for instance, a fluorescent protein that displays a redox-dependent change from green to blue [125]. Further, many studies reported the development of sensors for specific biomolecules that play an important role in the redox landscape, such as nicotinamide adenine dinucleotide (NADH) [126]. Furthermore, a recent study presented redox sensors for the cytosolic glutathione system, the major antioxidant system of the cell [127]. These genetically encoded redox proteins are very useful for monitoring the redox landscape under experimental conditions [128]. For instance, these sensors enable the study of molecular pathologies in animal models, such as the oxidative stress-mediated neurodegeneration of motor neurons in zebrafish [129]. Unfortunately, redox protein sensors depend on specific model systems, which prevents their application for diagnostic or therapeutic purposes in the human patient.

By contrast, redox sensors based on small molecules can be applied theoretically to humans, since the administration of compounds is not restricted to a defined model organism. Some recent examples include nicotinamide molecules that are conjugated to fluorophores, and serve as effective redox sensors in biological systems [130]. Other advances are multichannel redox sensors that combine chromogenic, fluorescent, and electrochemical signals for monitoring intracellular H_2_O_2_ in vivo [131]. However, small-molecule-based redox sensors require a specific chemical interaction between the compound and the target molecule; hence, their application is limited by the availability, selectivity, reversibility, and kinetics of the sensing reaction [123].

Besides the development of novel tools, another promising path for monitoring redox states in cellular model systems is the improvement of imaging devices. One recent example is the development of fluorescence lifetime imaging microscopy for single-cell ROS detection (FLIM–ROX) [132]. FLIM has been used before to detect the redox state in cells via reduction of NADH; however, the authors of this study applied FLIM in conjunction with selected ROS reporter dyes. The combination of the dye with the microscopic technique improved the sensitivity and reliability of ROS detection in lifetime imaging.

Research also explores some entirely novel approaches for monitoring oxidative stress in biological systems. A very recent and highly innovative approach to study the redox landscape in vivo in real-time was presented by Baltsavias et al. [133]. This group developed a small implant that is capable of measuring the balance of oxidants and reductants that are produced by the interplay of host, microbe, and diet in the gut. After implanting in rats, the device was powered externally with ultrasonic waves, and was able to monitor the redox state in the gut of the living animal for almost two weeks. This approach enables long-term experimental testing of redox pathophysiology mechanisms, and facilitates translation to disease diagnosis and treatment applications in the future.

### 3.2. Positron Emission Tomography (PET) for Monitoring Oxidative Stress

Noninvasive imaging methods provide diagnostic and prognostic information on functional states of the brain, and support effectivity assessment of new drug candidates [134,135]. Positron emission tomography (PET) is a molecular imaging modality, in which the administration of a positron emission radiotracer enables visualization and quantification of biochemical processes in vivo [136]. In neuroimaging, this radiotracer is often a small molecule that is labeled with fluorine-18 (half-life of 109.8 min) or carbon-11 (half-life of 20.3 min) radionuclides. An ideal PET radiotracer displays a high signal-to-noise ratio, which is defined as the ratio of specific to nonspecific binding, and increases its signal under a pathological versus a baseline (i.e., physiological) condition. Ideally, this signal also correlates with disease severity and responds to known therapeutic agents.

One proposed strategy for detecting ROS using PET relies on the change in cell permeability of dihydroethidium (DHT) tracers upon oxidation. A neutral fluorine-18 labelled DHT analog is BBB permeable, and becomes positively charged upon reacting with O_2_^•−^, trapping it intracellularly [137]. In a lipopolysaccharide (LPS) lesion model, this tracer showed enhanced accumulation in the ipsilateral side of the mouse brain, and the PET signal intensity correlated with the severity of the clinical symptoms [138]. Another family of O_2_^•−^ sensitive PET tracers is based on the dihydroisoquinoline scaffold. A carbon-11 labeled hydromethidine was initially described to accumulate in the rat brain after a LPS injury [139]. A fluorine-18 analog recently confirmed the retention of the tracer in the affected region [140]. However, this strategy detects only intracellular ROS, since the cationized tracer is only retained if formed inside the cell.

A notable study explored the differential uptake kinetics of oxidized versus reduced ascorbic acid, and overcame this limitation [141]. Ascorbic acid was labeled with carbon-11, and, after reacting with H_2_O_2_ or O_2_^•−^, it yielded [^11^C]dehydroascorbic acid ([^11^C]DHA), which is transported to cells ten times faster than ascorbic acid. Differential uptake of [^11^C]DHA was demonstrated in both cancer cell lines and also in human neutrophils, validating this approach. Another way of detecting extracellular ROS with PET focuses on the equilibrative nucleoside transporter 1 (ENT1). Here, the endogenous thymidine substrate was modified with a H_2_O_2_ carbamate-boronate-sensitive linker, and labeled with fluorine-18. Upon reaction with H_2_O_2_, the linker decomposes, making it a better substrate for ENT1-mediated internalization. Once inside the cell, the thymidine substrate cannot be further metabolized because of the fluorine-18 modification, which impairs enzyme recognition, eventually leading to intracellular tracer accumulation [142]. As a note, this tracer is also cell-permeable via passive diffusion and, hence, reports both intra- and extracellular ROS. The tracer signal responded to H_2_O_2_ dose-dependently, and was blocked by co-incubation with high concentrations of thymidine. However, the described applications were limited to cellular carcinoma models, perhaps due to the carbonate linker instability in vivo.

Only a few PET tracers were used in clinical studies to specifically probe redox processes in vivo. One of them is the copper chelator [^62/64^Cu]Cu-ATSM. After reacting with ROS, the reduced radioactive ^62/64^Cu^+^ dissociates from the complex, and is retained in the tissue [143]. This tracer was also applied in neurodegenerative diseases, such as Parkinson’s disease and amyotrophic lateral sclerosis (ALS), as well as in cancer [124,144]. Unlike the previously described tracers, which are reactive toward endogenous ROS, the [^18^F]fluoropropyl-glutamate tracer [^18^F]FSPG provides a measure of cellular antioxidant response via imaging a major contributor to the cellular redox homeostasis, i.e., the cystine/glutamate antiporter system x_C_^−^. [^18^F]FSPG is a substrate of system x_C_^−^, and cancer cells under oxidative stress upregulate this transporter, which is expressed at a low level in normal cells [145,146]. In a recent clinical trial, [^18^F]FSPG PET was used for the diagnosis of prostate cancer, confirming the high signal-to-background uptake levels [147]. Interestingly, this tracer was employed previously to detect the inflammatory component of the autoimmune disease sarcoidosis, even though it did not outperform the sugar analog [^18^F]FDG, which is routinely applied in PET [148].

## 4. Imaging Redox Biomarkers of Neuroinflammation in the Clinic

Several biomarkers and associated radiotracers were proposed for the detection of different steps in the neuroinflammation biochemical cascade. We refer the interested reader to two detailed and comprehensive overviews of recent neuroinflammation PET biomarkers [149,150]. Here, we provide a brief overview of two imaging biomarkers for neuroinflammation, which are linked to the redox landscape: TSPO and MAO–B (Figure 3). Both imaging biomarkers play an important role in the cellular redox landscape, and are associated with clinically validated tracers. TSPO is involved in mitochondrial activity, whereas MAO–B generates H_2_O_2_ in the mitochondria [151,152]. As of now, PET is one of the few methods that warrants adequate sensitivity for measuring the abundance of transiently expressed redox-related biomarkers of neuroinflammation with translational potential.

### 4.1. Translocator Protein (TSPO)

TSPO, also known as peripheral benzodiazepine receptor (PBR), is an abundantly expressed protein that is mainly located in the mitochondria, where it acts as a cholesterol transporter protein [153]. Notably, TSPO plays an important role in the amplification of ROS in the mitochondria [151,154]. The expression of TSPO in microglia is significantly upregulated upon injury or inflammation in the brain [155]. Since basal TSPO expression in the CNS is low, its upregulation indicates CNS infiltration and activation of immune cells in the brain [156]. TSPO was the first neuroinflammation biomarker for PET radiotracers, and it is still the most explored target.

Currently, three generations of TSPO PET ligands exist. The first contains the carbon-11-labeled isoquinoline [^11^C]PK11195. This molecule was discovered in the early 1980s and was described as binding to a peripheral benzodiazepine-binding site of the γ-aminobutyric acid (GABA) receptor, which corresponds to TSPO, as later studies demonstrated [157,158]. Early preclinical reports showed that racemic PK11195 binds in ischemic lesions in rats, and that the *R*-isomer tracer binding is 2-fold higher than the *S*-analog [159]. A subsequent clinical study using the *R*-enantiomer of PK11195 demonstrated that high levels of tracer accumulated at pathological sites in MS and, according to magnetic resonance imaging (MRI), in normal-appearing white-matter regions, as well [160]. This study also explained the *R*-[^11^C]PK11195 signal increase by a high number of binding sites, and excluded the possibility of an altered affinity toward TSPO. Nonetheless, additional studies indicated poor pharmacokinetics, low brain uptake, and high nonspecific binding levels, which still poses a challenge for quantification and modeling of *R*-[^11^C]PK11195 uptake in pathological brain areas [161].

A second-generation of TSPO ligands was developed, including pyrazolopyrimidine tracers labeled with fluorine-18, and eventually compared with [^11^C]PK11195 in a rat model of neuroinflammation. For example, [^18^F]DPA-714 accumulation in the ipsilateral side to the lesion was five-times higher than *R*-[^11^C]PK11195, and demonstrated lower nonspecific binding [162]. In 2012, the discovery of a single nucleotide mutation in the human *TSPO* gene finally explained the high inter-individual variability that clinical trials with second-generation TSPO ligands had observed previously [163]. Patient stratification prior to the clinical trials may overcome this issue; however, this observed genetic variability still limits the application of TSPO as a biomarker for neuroinflammation and hampers patient enrollment.

Currently, efforts are underway in designing and developing a third generation of TSPO ligands that specifically address the differential response due to *TSPO* polymorphisms [164]. Nevertheless, it is still unclear whether these compounds will match the requirements that are needed for irrevocable radioligands for PET imaging of activated microglia.

Despite the challenges of TSPO imaging, numerous clinical trials confirmed that all three generations of TSPO PET tracer are able to delineate brain neuroinflammation [165,166]. TSPO imaging has been used to track disease severity and the efficacy of treatments in neurodegenerative disorders, such as Alzheimer’s disease and dementia [167,168]. A recent study showed a reduction in binding of the first-generation *R*-[^11^C]PK11195 tracer in the brain of patients with MS after treatment with the therapeutic antibody Natalizumab [169]. Importantly, high PET signals in white-matter regions correlated with more rapidly progressing disease after four years of imaging, which indicates the prognostic value of this tracer. Another clinical study on MS demonstrated that binding of the second-generation tracer [^11^C]PBR28 correlated with deteriorating cognitive performance and neurological scores [170]. This study further highlighted the benefits of PET imaging, in particular, since morphologically normal-appearing white matter produced abnormally high [^11^C]PBR28 signal.

Even though TSPO imaging of neuroinflammation has not yet reached routine clinical applications, an extensive number of clinical studies confirmed the correlation of TSPO upregulation with in vivo glial activation, disease symptom severity, and, in some cases, with disease prognosis.

### 4.2. Monoamine Oxidase B (MAO–B)

Despite the progress on TSPO ligands over the years, the work on preclinical models pointed out the advantages of other neuroinflammation biomarkers in diseases, such as AD and MS [171]. One of these proposed new targets is the enzyme MAO–B.

MAO–B is expressed in mitochondria, and is an important component of the cellular redox landscape in the brain [172,173]. The enzyme produces H_2_O_2_, modulates oxidative stress, and, hence, contributes to inflammation [174]. In the brain, MAO–B is located mainly in the outer mitochondrial membrane of astrocytes and neurons [175]. Certain diseases, such as AD or ALS, are associated with an increased number of activated astrocytes, which correlates with MAO–B expression. Overexpressed MAO–B also colocalizes with the astrocyte marker glial fibrillary acidic protein (GFAP).

Molecular imaging agents that target MAO–B have been known since the late-1980s [176]. The first class of PET tracers are based on suicide inhibitors that contain an acetylenic residue for binding to the flavin group at the active site of MAO–B [177]. In the brain, the signal from L-[^11^C]deprenyl and its more recent deuterated form L-[^11^C]deprenyl-D_2_ is sensitive to pharmacological modulation, which correlates reproducibly with the distribution of MAO–B in humans [178,179,180]. However, tracer modeling is challenging, since this class of ligands is metabolically converted to methamphetamine, which is brain-penetrant and pharmacologically active [181,182].

Novel ligands, such as the oxazolidinone [^11^C]SL25.1188, are promising tracers with reversible and high-affinity binding for MAO–B [183]. In baboons, [^11^C]SL25.1188 showed no formation of brain-penetrant metabolites, was eliminated faster than L-[^11^C]deprenyl-D_2_, and its signal correlated with MAO–B distribution [184]. Kinetic modeling of [^11^C]SL25.1188 confirmed the favorable pharmacokinetics and paved the way for this tracer as an imaging biomarker for MAO–B [185]. As of now, the disease relevance of [^11^C]SL25.1188 has been studied only in patients with major depressive disorder (MDD) [186]. In these studies, the tracer levels were 25 % higher in the prefrontal cortex of MDD patients than in healthy controls. This suggests a new pathophysiological hypothesis for depression that involves mitochondrial dysfunction or monoamine metabolism imbalance.

### 4.3. Recent Examples of PET Imaging of Neuroinflammatory Biomarkers in Stroke and Alzheimer’s Disease

Ischemic stroke pathology entails a very dynamic and acute neuroinflammatory component with multiple molecular hallmarks within different parts of the brain. Currently, imaging methods pinpoint the location, extension, and type of the stroke lesion, and they estimate penumbral regions, i.e., potentially salvageable brain regions surrounding the primary lesion [187]. TSPO PET imaging revealed a slow increase in tracer uptake localized in the periphery of the main lesion until the third day after stroke. Then, a sudden rise in TSPO signal occurs and peaks after two weeks, also spreading into the core lesion. After this time, the tracer levels at the ischemic core decline but reach later areas remote from the initial stroke site [188]. Neuroinflammation imaging may guide and tailor therapeutics to timely induce inflammatory tissue repair or to reduce global inflammatory mechanisms [189].

In the case of AD, PET molecular imaging has already contributed to the diagnostics of two classical hallmarks: Aβ plaques and neurofibrillary tangles (NFTs) [190]. PET molecular imaging also provided compelling evidence for the presence of an underlying chronic inflammatory component (reviewed in [191] and [192]) of AD. A multitracer study tracked the associated inflammatory processes, and shed light on the complex pathophysiology of AD. *R*-[^11^C]PK11195, combined with tracers for Aβ and tau protein, revealed a biphasic neuroinflammation pattern in AD patients: initially, the *R*-[^11^C]PK11195 signal was high in subjects bearing low Aβ levels and mild cognitive impairment (MCI) but, eventually, *R*-[^11^C]PK11195 levels decreased as Aβ loads rose to pathological levels. At this later phase, the tau protein levels increased, and prompted the second wave of microglial activation, reaching the pathological phenotype of concomitant high levels of Aβ and tau protein and cognitive deterioration [193]. Similarly, the signal of the second-generation TSPO tracer [^11^C]PBR28 was elevated in more than 50 MCI patients, potentially indicating a protective microglial activation at early AD [194]. In a remarkable longitudinal study, patients with genetic predisposition for AD were scanned for both Aβ and MAO–B using L-[^11^C]deprenyl-D_2_. The images showed high levels of astrocytosis in asymptomatic subjects initially, which steadily declined over the course of two to three years. Moreover, the PET signal was also elevated in patients carrying autosomal-dominant AD mutations, even seventeen years before the expected onset of symptoms, which provides a window for AD early diagnosis [195].

## 5. Conclusions

A growing body of research provides evidence that inflammation in the brain is a driving force in the development of many human diseases. Research on the molecular events of brain inflammation, in particular its redox component, is a challenging endeavor, due to the high complexity of the inflammatory process and the brief lifetime of redox molecules. In addition, the seclusion of the brain (and the CNS) from the blood circulation by the BBB hampers the discovery of biomarkers and the development of novel therapeutic compounds for effective therapeutic intervention. Furthermore, the lack of tools to monitor and visualize the dynamic states of brain inflammation in the patient accounts for inadequate diagnostic options in many neurological disorders.

The recent advent of chemical biology provides medical professionals with the proper means for diagnostics, and scientists with an increasing arsenal of tools to elucidate the underlying molecular basis of inflammatory brain diseases. Chemical biology and diagnostic tools will further push the boundaries of our understanding and the success of medical interventions. Given the increasing incidence of inflammatory brain diseases, such as stroke and AD, these developments are urgently needed.

## Figures and Tables

**Figure 1 antioxidants-10-00528-f001:**
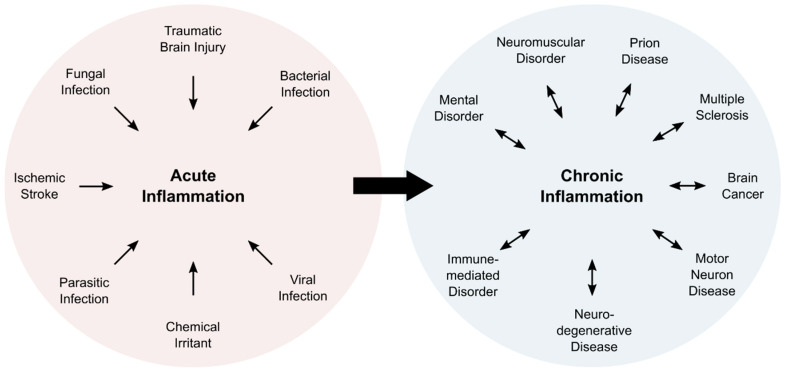
Overview of cause and consequence of acute and chronic inflammation. Acute inflammation is induced in response to external factors. Persisting inflammatory responses become chronic events that are associated with numerous pathologies.

**Figure 2 antioxidants-10-00528-f002:**
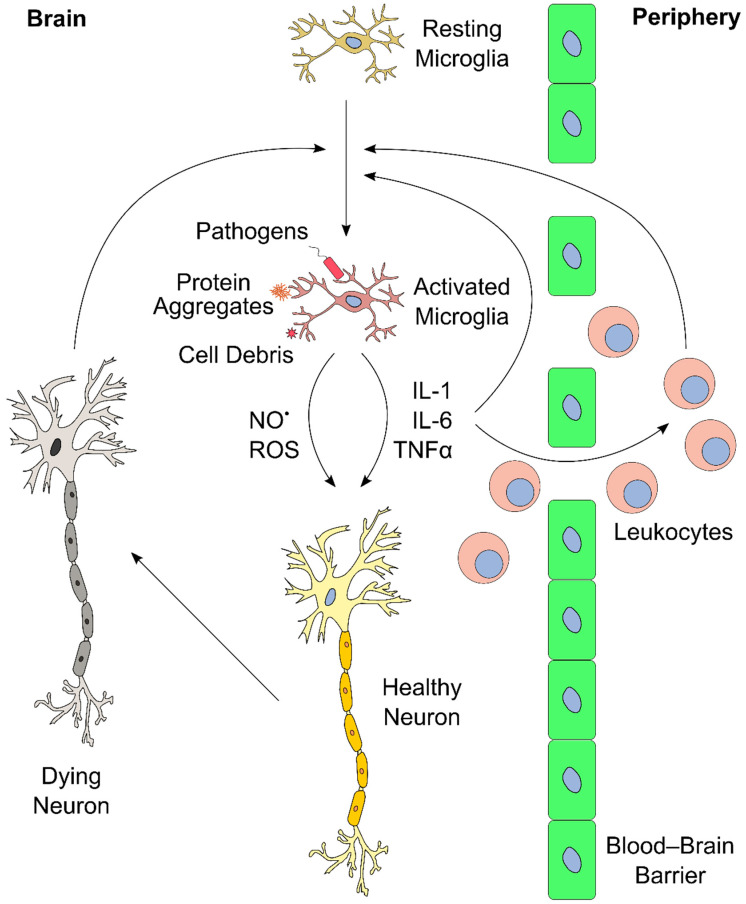
Schematic overview of brain inflammation. Resting microglia in the brain become activated upon exposure to external stimuli (e.g., pathogens, protein aggregates, cell debris). This induces proliferation and release of cytokines (e.g., interleukin-1 (IL-1), IL-6, and tumor necrosis factor α (TNF-α)), reactive oxygen species (ROS), and reactive nitrogen species (RNS, such as NO^•^). These inflammatory mediators further stimulate other microglia in the brain. In addition, cytokines and ROS affect neurons, and propagate cell death upon extended exposure, which, in turn, could further stimulate microglia in the central nervous system (CNS). Furthermore, the inflammatory mediators act in the blood–brain barrier, and recruit leukocytes from the periphery into the brain. Leukocytes can perpetuate the inflammatory process even further, for instance, via secretion of additional cytokines. These positive feedback mechanisms can lead to a vicious cycle of inflammation and neuronal death.

**Figure 3 antioxidants-10-00528-f003:**
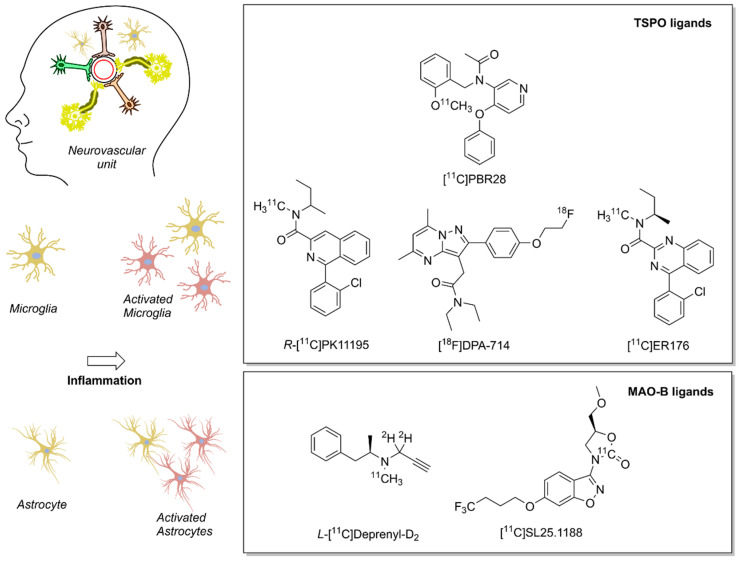
Positron emission tomography (PET) radioligands for translational molecular imaging of two biomarkers of neuroinflammation. The 18 kDa translocator protein (TSPO) and the enzyme monoamine oxidase B (MAO–B) are overexpressed in response to inflammation predominately in microglia and astrocytes, respectively.

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
