# Peer review of "Imaging Biomarkers for Monitoring the Inflammatory Redox Landscape in the Brain"

_antioxidants, 2021, doi:10.3390/antiox10040528_

Round 1

Reviewer 1 Report

The review “Imaging Biomarkers for Monitoring the Infammatory Redox Landscape in the Brain” by Fernandes and Özcelik is a well-written, structured and concise overview of several imaging and detection methods of oxidative stress in biological systems, both in vitro and in vivo. I have some minor remarks related to the PET part (section 4) of this review article, please see below.

Authors state  that neuroinflammatory response fuels oxidative stress – yet it is not clear to me why only two PET imaging targets of the neuroinflammatory response are mentioned and specifically why these two. Besides TSPO and MAO-B a number of newer targets have been investigated, even in the clinic. Given the scope of the journal it is understandable not all of these are mentioned, but as this has been extensively reviewed over the past years, I would advise to refer the reader to a recent neuroinflammation PET review article with a more complete overview of all neuroinflammation targets.

There’s quite a few [11C]PK11195 occurrences that have an extra 5 at the end. Please check the entire manuscript thoroughly to correct this. In addition, most likely the described studies were performed with the R-isomer rather than the racemic tracer, please add this for each study as it could have influenced study outcome.

Please be consistent in using square brackets for radioactive isotopes in PET tracers as this is according to nomenclature guidelines (e.g. [11C]PK11195 and [18F]DPA-714 rather than 11C-PK11195 and 18F-DPA-714).

Author Response

* We would like to thank the reviewer for the helpful comments. We appreciate the generally positive perception of our work, and we gladly incorporated the provided suggestions in order to improve our manuscript.

The review “Imaging Biomarkers for Monitoring the Infammatory Redox Landscape in the Brain” by Fernandes and Özcelik is a well-written, structured and concise overview of several imaging and detection methods of oxidative stress in biological systems, both in vitro and in vivo. I have some minor remarks related to the PET part (section 4) of this review article, please see below.

Authors state  that neuroinflammatory response fuels oxidative stress – yet it is not clear to me why only two PET imaging targets of the neuroinflammatory response are mentioned and specifically why these two.

Besides TSPO and MAO-B a number of newer targets have been investigated, even in the clinic. Given the scope of the journal it is understandable not all of these are mentioned, but as this has been extensively reviewed over the past years, I would advise to refer the reader to a recent neuroinflammation PET review article with a more complete overview of all neuroinflammation targets.

* We thank the reviewer for the positive assessment of our work. Indeed, multiple targets and associated tracers have been proposed for the detection of neuroinflammation. We decided to limit the number of examples to TSPO and MAO-B, since both tracers have been extensively studied and validated for imaging inflammatory responses. Moreover, both biomarkers have a direct or indirect relationship with redox processes, which is an important aspect of this review. In addition, both proteins play a role in inflammasome signaling pathway. We described this rationale at the beginning of the new section 2.2 (see line 166-176), and at the beginning of section 4 (see lines 362-368). Furthermore, we referenced two recent reviews on PET imaging biomarkers for neuroinflammation to refer the reader towards more comprehensive reviews on the topic, as suggested (see lines 362-264).

There’s quite a few [11C]PK11195 occurrences that have an extra 5 at the end. Please check the entire manuscript thoroughly to correct this. In addition, most likely the described studies were performed with the R-isomer rather than the racemic tracer, please add this for each study as it could have influenced study outcome.Please be consistent in using square brackets for radioactive isotopes in PET tracers as this is according to nomenclature guidelines (e.g. [11C]PK11195 and [18F]DPA-714 rather than 11C-PK11195 and 18F-DPA-714).

* We thank the reviewer for pointing out this issue. We revised all the instances where [11C]PK11195 was misspelled, and indicated which enantiomer was used in the referenced studies. Additionally, we adopted the recommended radiochemistry nomenclature for all mentioned radioligands in the main text and in Figure 3, according to the following reference:

Coenen, H.H.; Gee, A.D.; Adam, M.; Antoni, G.; Cutler, C.S.; Fujibayashi, Y.; Jeong, J.M.; Mach, R.H.; Mindt, T.L.; Pike, V.W.; et al. Open letter to journal editors on: International Consensus Radiochemistry Nomenclature Guidelines. EJNMMI Radiopharmacy and Chemistry 2019, 4, 7, doi:10.1186/s41181-018-0047-y.

Reviewer 2 Report

This report is timely and extremely relevant and will deepen our understanding of the biomarkers to monitor neuroinflammation in brain. The manuscript was well written. The authors should include the below information to improve the manuscript before reconsidering for publication.

  1. The authors should provide more information on disease specific neuroinflammatory events in Alzheimer's disease, multiple sclerosis and Huntington's disease.
  2. A separate section on immune cells, inflammasome and immune cell activation should be provided.
  3. Include additional information on inflammatory pathways of brain. 
  4. A brief information on targeting neuroinflammation should be added.

Author Response

* We are glad that the reviewer recognizes the value of our review. We appreciate the helpful comments and suggestions from the reviewer, and we addressed all points in order to improve our manuscript.

This report is timely and extremely relevant and will deepen our understanding of the biomarkers to monitor neuroinflammation in brain. The manuscript was well written. The authors should include the below information to improve the manuscript before reconsidering for publication.

  1. The authors should provide more information on disease specific neuroinflammatory events in Alzheimer's disease, multiple sclerosis and Huntington's disease.

* We thank the reviewer for this suggestion. We provided more information on neuroinflammatory events in Alzheimer’s disease, in particular the role of IFITM3 and TREM-2 (see lines 44-53). Of course, we cited the relevant literature accordingly. AD is an important example for chronic neuroinflammation but neurodegenerative diseases are not the focus of this review; hence, we referred the interested reader to an up-to-date review on this topic.

In keeping with the focus of this review, we included only a few additional details on the role of inflammation in multiple sclerosis (see lines 35-38). Since we never mentioned Huntington’s disease in this review, we decided not to include new information on this indication.

With the expansion on Alzheimer’s disease, we believe that we have already given sufficient context on inflammation in neurodegenerative diseases. In our opinion, adding another one, we would shift the focus of this review too much towards this type of pathology. We think that this shift in focus would then raise the question, why we made this choice, and why we neglected other important neuroinflammatory disorders, such as spinal muscular atrophy, depression, traumatic brain injury, viral encephalitis, or autism spectrum disorder. Given the limitation of the length and scope of this review, we hope the reviewer will not hold this limited selection against us.

  1. A separate section on immune cells, inflammasome and immune cell activation should be provided.

* We thank the reviewer for this valuable suggestion to our manuscript, and we agree that such a section will improve our review. Hence, we added a new section to the manuscript, i.e. “2.2. Induction of the inflammatory response” (line 152ff). Here, we added information on immune cell activation, followed by a paragraph linking immune cell activation to the redox landscape to maintain the focus of this review. Further, we also included a brief description on the inflammasome and the involved immune cells, as the reviewer suggested.

  1. Include additional information on inflammatory pathways of brain.

* We appreciate the suggestion of the reviewer. With the addition of section 2.2 and the description of the role of the inflammasome, we believe that we have provided sufficient examples of relevant inflammatory pathways of brain. Nonetheless, we provided additional and specific information in section 2.3 on the occurrence of pattern recognition receptors (PRRs) in neuronal cells, including a detailed set of references (see lines 204-207).

  1. A brief information on targeting neuroinflammation should be added.

* The reviewer made a very good point. We acknowledge the importance of pharmacological modulation of neuroinflammation, and although therapeutics are not the focus of this review, we provided brief information on targeting inflammatory pathways at the end of section 2.2, and cited four reviews on this topic (see lines 188-196).

Round 2

Reviewer 2 Report

The manuscript was improved substantially by revision and may be accepted for publication.